# Ultrastructural effects of sleep and wake on the parallel fiber synapses of the cerebellum

Sophia S Loschky[1†], Giovanna Maria Spano[1†], William Marshall[1,2], Andrea Schroeder[1], Kelsey Marie Nemec[1], Shannon Sandra Schiereck[1], Luisa de Vivo[1], Michele Bellesi[1], Sebastian Weyn Banningh[1], Giulio Tononi[1], Chiara Cirelli[1]*

[1]Department of Psychiatry, University of Wisconsin-Madison, Madison, United States; [2]Department of Mathematics and Statistics, Brock University, St. Catharines, Canada

*For correspondence: ccirelli@wisc.edu

†These authors contributed equally to this work

Competing interest: The authors declare that no competing interests exist.

**Abstract** Multiple evidence in rodents shows that the strength of excitatory synapses in the cerebral cortex and hippocampus is greater after wake than after sleep. The widespread synaptic weakening afforded by sleep is believed to keep the cost of synaptic activity under control, promote memory consolidation, and prevent synaptic saturation, thus preserving the brain's ability to learn day after day. The cerebellum is highly plastic and the Purkinje cells, the sole output neurons of the cerebellar cortex, are endowed with a staggering number of excitatory parallel fiber synapses. However, whether these synapses are affected by sleep and wake is unknown. Here, we used serial block face scanning electron microscopy to obtain the full 3D reconstruction of more than 7000 spines and their parallel fiber synapses in the mouse posterior vermis. This analysis was done in mice whose cortical and hippocampal synapses were previously measured, revealing that average synaptic size was lower after sleep compared to wake with no major changes in synapse number. Here, instead, we find that while the average size of parallel fiber synapses does not change, the number of branched synapses is reduced in half after sleep compared to after wake, corresponding to ~16% of all spines after wake and ~8% after sleep. Branched synapses are harbored by two or more spines sharing the same neck and, as also shown here, are almost always contacted by different parallel fibers. These findings suggest that during wake, coincidences of firing over parallel fibers may translate into the formation of synapses converging on the same branched spine, which may be especially effective in driving Purkinje cells to fire. By contrast, sleep may promote the off-line pruning of branched synapses that were formed due to spurious coincidences.

## Editor's evaluation

This study provides compelling structural evidence on regulation of cerebellar synapses by sleep-wake states. The authors used serial block face scanning electron microscopy to obtain 3D reconstruction of more than 7,000 spines and their parallel fiber synapses in the mouse posterior vermis. The analysis shows that sleep increases the fraction of the 'naked' spines that don't carry a presynaptic partner at Purkinje cells and the authors propose that sleep promotes the pruning of branched synapses to single spines. This is an elegant and thorough study and the observations are important in light of the circuit-specific mechanisms by which sleep modulate synaptic structure and function.

## Introduction

Converging evidence in rats and mice shows that many cortical and hippocampal excitatory synapses are stronger after wake than after sleep (*Cirelli and Tononi, 2021*). Excitatory synaptic strength is thought to increase on average during wake due to incidental learning leading to net synaptic potentiation, and to decrease during sleep due to synaptic renormalization that is both widespread and selective (*Tononi and Cirelli, 2014*; *Cirelli and Tononi, 2021*). The net decrease in synaptic strength afforded by sleep is likely to serve an essential function, maintaining the energetic cost of synaptic activity under control, promoting memory consolidation, and preserving the brain's ability to learn by avoiding synaptic saturation (*Diekelmann and Born, 2010*; *Tononi and Cirelli, 2014*).

In rodents the evidence for an overall increase in synaptic strength during wake and a net decrease during sleep comes from multiple sources. Electrophysiologically, experiments in mice and rats found that the slope and amplitude of cortical evoked responses, as well as the amplitude and/or frequency of miniature excitatory postsynaptic currents, are higher after wake than after sleep (*Vyazovskiy et al., 2008*; *Liu et al., 2010*; *Bjorness et al., 2020*; *Bridi et al., 2020*; *Khlghatyan et al., 2020*). The molecular evidence was provided by studies in synapse-rich preparations obtained from cortex, hippocampus, and the whole forebrain. These experiments showed that the expression of the glutamatergic AMPA receptors is higher after several hours of sustained wake relative to several hours of consolidated sleep, independent of whether the animals were spontaneously awake at night or sleep deprived during the day (*Vyazovskiy et al., 2008*; *Diering et al., 2017*; *Miyamoto et al., 2021*).

Ultrastructural evidence has been obtained using serial block face scanning electron microscopy (SBEM) (*Cirelli and Tononi, 2020*). It was found that the average size of the axon-spine interface (ASI), a structural measure of synaptic strength (*Desmond and Levy, 1988*; *Buchs and Muller, 1996*; *Cheetham et al., 2014*), is larger after wake than after sleep in the axospinous synapses of the mouse primary motor and sensory cortex (*de Vivo et al., 2017*), as well as in the hippocampal CA1 region (*Spano et al., 2019*). By contrast, synapse number does not change between sleep and wake except for the nonperforated synapses of CA1, whose density increases after sleep deprivation (*Spano et al., 2019*). Thus, the effects of sleep and wake in cortex and hippocampus are not so much due to changes in synapse number, but to changes in the surface expression, subunit composition, and

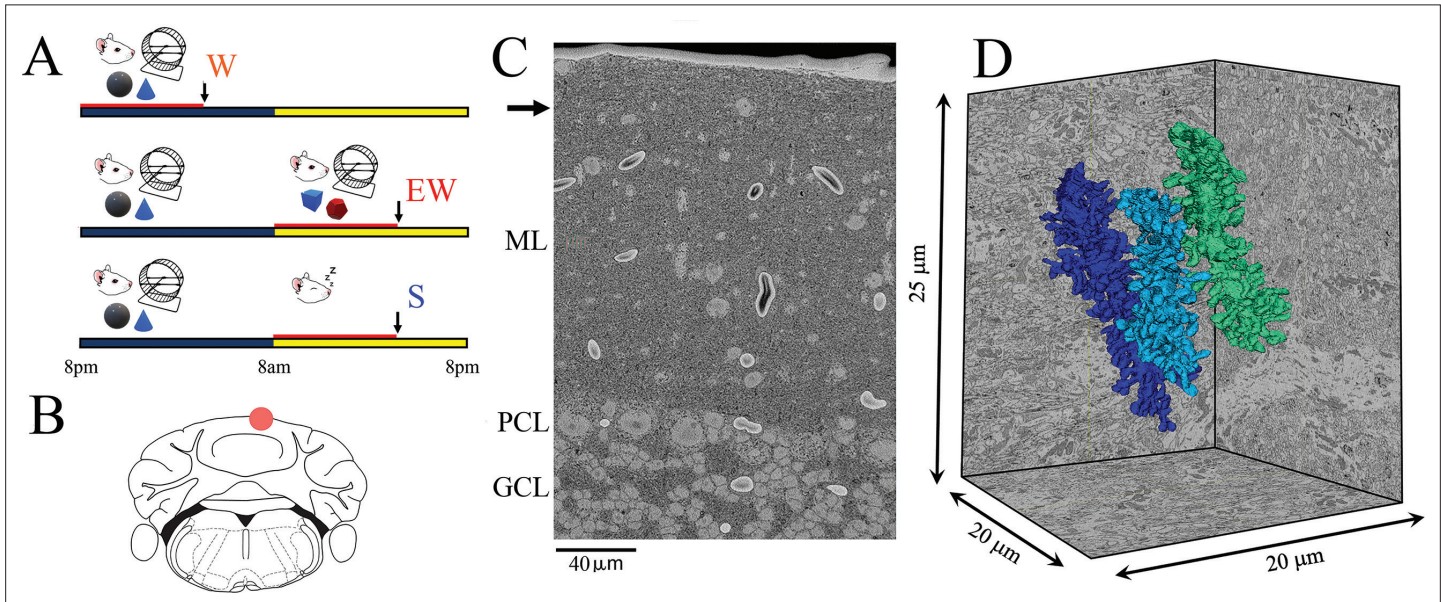

**Figure 1.** Experimental design. (**A**) The three experimental groups. W, spontaneous wake at night; EW, extended wake during the day; S, sleep during the day. Black and yellow horizontal bars indicate the 12 hr dark and 12 hr light phases, respectively. The red horizontal bar marks the last 6–8 hr before the time of brain collection (black arrow). (**B**) Schematic representation of a coronal section of the mouse brain with the area of the posterior vermis (lobule VI); the area used for serial block face scanning electron microscopy (SBEM) imaging is indicated by a red circle. (**C**) Low magnification image showing the molecular layer (ML), Purkinje cell layer (PCL), and granule cell layer (GCL). The black arrow indicates the area where the image stack was collected. (**D**) 3D reconstruction of three dendritic branches in one stack.

phosphorylation levels of the AMPA receptors, which ultimately control the trafficking of these receptors in and out of the synaptic membrane (*Cirelli and Tononi, 2021*).

In the mammalian brain the great majority of neurons and synapses are housed in the cerebellum, which remains highly plastic even during adulthood (*Imamizu et al., 2000*; *Hansel et al., 2001*). Cerebellar granule cells account for approximately 60 out of 86 billions in the human brain (*Herculano-Houzel, 2010*) and the glutamatergic parallel fiber synapse, between the parallel fibers of the granule cells and the spiny dendrites of the Purkinje cells, is the most abundant type of cerebellar synapse, with each Purkinje cell in the rat receiving ~175,000 parallel fiber inputs (*Napper and Harvey, 1988a*). These synapses undergo long-term depression in response to the paired stimulation of parallel fibers and climbing fibers, a form of plasticity that has classically been linked to motor learning (*Ito, 2001*). Presynaptic and postsynaptic forms of long-term potentiation are also present at these and other cerebellar synapses and contribute to different forms of learning (*Gao et al., 2012*; *D'Angelo et al., 2016*; *Romano et al., 2018*). However, very little if anything is known about how sleep and wake modulate cerebellar plasticity in general (*Canto et al., 2017*) and, more specifically, whether they affect the number and strength of the parallel fiber synapses. Here, we used SBEM to investigate whether the parallel fiber synapses of the mouse cerebellar posterior vermis undergo ultrastructural changes after several hours of sleep, spontaneous wake, and sleep deprivation.

## Results

Three groups of mice were selected based on the sleep/wake behavior during the last 6–8 hr before the brains were collected, according to strict criteria as in previous studies (see Materials and methods for details). They included a spontaneous waking (W) group of mice that were mostly awake for the first 6–8 hr of the dark period (n=4), an extended waking (EW) group of animals that were sleep deprived during the first 6–8 hr of the light period (n=4), and a sleep (S) group of mice that spent most of the first 6–8 hr of the light phase asleep (n=6; *Figure 1A*). These 14 mice were the same used to assess ultrastructural sleep/wake synaptic changes in CA1 (*Spano et al., 2019*), and 8 of them were also used for the analysis of primary motor and sensory cortex (*de Vivo et al., 2017*). As previously reported (*Spano et al., 2019*), in the last 6–8 hr W mice were awake for ~86% of the time, EW mice were kept continuously awake, and S mice slept ~84% of the time (*Spano et al., 2019*). The S mice were compared to both wake groups to identify ultrastructural changes driven by the sleep/wake behavior while controlling for time-of-day effects (day vs. night) and the effects of sleep deprivation (spontaneous vs. forced wake) (*Figure 1A*).

In each mouse we used SBEM to acquire stacks of ~500 images (~10,000 μm$^3$) in the posterior vermis of lobule VI (*Figure 1B*), a region that contains mostly zebrin II positive Purkinje cells (*Cerminara et al., 2015*). We specifically targeted the superficial part of the molecular layer, approximately 20–30 μm from the pial surface (*Figure 1C*), where the distal tips of the dendritic branchlets are located (*Ichikawa et al., 2016*). Our 3D reconstructions showed that roughly two-thirds of the segmented dendrites (66 out of 96) were terminal branches (*Figures 1D and 2*). In these dendritic tips most synapses are established by the parallel fibers, with few if any climbing fiber synapses (*Ichikawa et al., 2016*). Spiny dendritic segments were randomly selected within each stack and all their protrusions, also called spines (see Materials and methods), were manually segmented by trained annotators blind to experimental condition. The length and diameter of the dendrites were balanced across groups (see Materials and methods; *Table 1*). Overall, across all mice 96 dendritic branches were segmented (S=43; W=29; EW = 24) (*Figure 2*) for a total of 7388 spines. Of these spines, 6853 had a synapse, and in 6618 of them the ASI could be fully traced and measured (N of ASIs, S=3004; W=1766; EW = 1848; at least 358 ASIs/mouse) (*Table 2*).

### Density of spines and synapses

All dendritic branches were very spiny (*Figures 2 and 3A*) with an average spine density across all mice of around 2 per dendrite surface area (*Table 2*). The great majority of these spines (92.7% across all mice) contained a synapse and in almost all cases only one synapse was present in each spine, in line with previous studies (*Harris and Stevens, 1988*; *Napper and Harvey, 1988a*; *Napper and Harvey, 1988b* ). First, we tested for sleep/wake effects on the absolute density of all spines and separately for spines with and without a synapse. By applying linear mixed effect (LME) models with

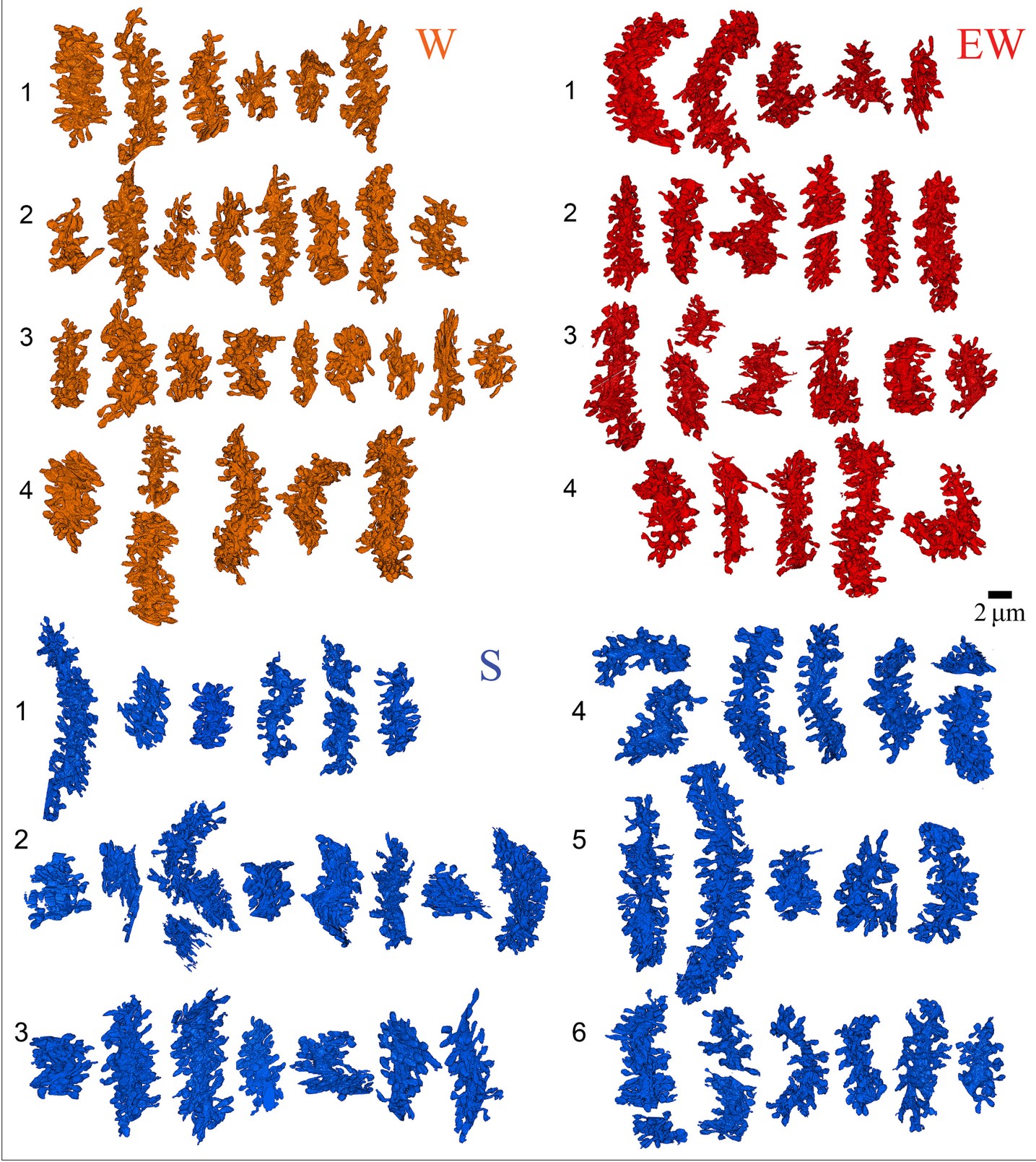

**Figure 2.** Reconstruction of all dendritic segments used in the study. W, spontaneous wake (4 mice, 29 dendrites); EW, extended wake (4 mice, 24 dendrites); S, sleep (6 mice, 43 dendrites). Each mouse is identified by a number.

**Table 1.** Parameter estimates for the linear mixed effect (LME) models used to assess changes in dendrite diameter and length.

Residual plots showed no evidence against the assumptions of constant variance and normality.

| LME model – dendrite diameter | | | |
|---|---|---|---|
| **Random effects** | **Standard error** | | |
| Mouse (intercept) | 0.0680 | | |
| Residual | 0.1500 | | |
| **Fixed effects** | **Level** | **Estimate** | **Standard error** |
| Intercept | | 1.3133 | 0.0459 |
| Condition | EW (reference) | 0 | 0 |
| | W | –0.0024 | 0.0637 |
| | S | 0.0698 | 0.0584 |
| **LME model – dendrite length** | | | |
| **Random effects** | **Standard error** | | |
| Mouse (intercept) | 0.8024 | | |
| Residual | 3.2883 | | |
| **Fixed effects** | **Level** | **Estimate** | **Standard error** |
| Intercept | | 8.7228 | 0.7841 |
| Condition | EW (reference) | 0 | 0 |
| | W | –0.4763 | 1.0734 |
| | S | 0.0007 | 0.9878 |

condition as a fixed effect and mouse as a random intercept (Materials and methods, *Table 3*), we found that the absolute density of spines did not change across experimental groups (p=0.1774; N of spines per dendrite surface area; mean ± std; W=1.83 ± 0.30; EW = 2.16 ± 0.44; S=1.97 ± 0.51) (*Figure 3B*). The absolute density of synapses also did not change across experimental groups (p=0.1476; N of synapses per dendrite surface area; mean ± std; W=1.76 ± 0.28; EW = 2.04 ± 0.41; S=1.78 ± 0.46) (*Figure 3C* and *Table 2*). On the other hand, the density of spines without a synapse ('naked' spines), which account for ~7% of all spines across mice, showed a significant effect of condition (p=0.0023; mean ± std; W=0.08 ± 0.06; EW = 0.13 ± 0.07; S=0.19 ± 0.10). Post hoc tests found a significant difference between S and W mice, with the observed density of spines without a synapse being 58% higher in the S group relative to the W group (p=3e-5; *Figure 3D*). A similar trend was present between S and EW mice (23% higher in S; p=0.0906; *Figure 3D*). Using an LME model to test for changes in the proportion of spines with and without a synapse relative to all spines, we also found a significant effect of condition (p=0.0014): in S mice the spines lacking a synapse represented ~10% of all spines, roughly double the proportion in W (p=2e-5) and EW (p=0.0138) mice (*Figure 3E*, left). Conversely, spines with a synapse represented ~95% of all spines in W and EW mice and significantly less, ~90%, in S mice (S<W, p=0.0001; S<EW, p=0.0139; *Figure 3E*, right). There were several types of spines without a synapse, including single spines (*Figure 3F*), branched spines with two spines both lacking a synapse (*Figure 3G*), branched spines with one spine with and the other without a synapse (*Figure 3H*), and spines lacking a synapse coming off the head of a spine with a synapse (*Figure 3I*). S mice showed a higher proportion of spines without a synapse of all types, but only in a few cases the comparison with the wake groups was statistically significant (*Figure 3J–M*).

Overall, these results show that the proportion of spines lacking a synapse is higher after sleep than after wake while the opposite is true for the spines that harbor a synapse, and both single and branched spines likely contribute to these differences. These changes occur without net changes in the total number of spines or synapses, most likely because spines lacking a synapse account for a small minority of all spines.

**Table 2.** Summary of ultrastructural measures in the molecular layer of the cerebellum. Previously reported measures in the hippocampus CA1 region (**Spano et al., 2019**) and cerebral cortex (**de Vivo et al., 2017**) are also shown to facilitate the comparison across regions. All protrusions are defined as spines. In oblique spines the axon-spine interface (ASI) could not be measured because the synapse was oriented obliquely or orthogonally to the cutting plane. Standard deviations for the cortex use a pooled estimate based on two sampled regions. MVB, multivesicular body.

| | CA1 (stratum radiatum) | | | Cortex (M1+S1, layer 2) | | | Cerebellum | | |
|---|---|---|---|---|---|---|---|---|---|
| | W | EW | S | W | EW | S | W | EW | S |
| Total N of dendrites | 37 | 26 | 38 | 54 | 57 | 57 | 29 | 24 | 43 |
| Total N of spines | 2764 | 2139 | 3148 | 2919 | 2616 | 2892 | 1944 | 2050 | 3394 |
| Total N of spines with synapse (some incomplete/oblique) | 2702 | 2080 | 3037 | 2467 | 2267 | 2415 | 1861 | 1926 | 3066 |
| Total N spines with synapse with measured ASI | 2552 | 1949 | 2840 | 2180 | 1989 | 2136 | 1766 | 1848 | 3004 |
| ASI (µm², mean ± std) range (µm²) | 0.128±0.122 0.003–1.033 | 0.130±0.122 0.008–0.764 | 0.120±0.108 0.006–0.672 | 0.299±0.323 0.012–4.019 | 0.290±0.305 0.005–3.543 | 0.248±0.278 0.006–2.038 | 0.187±0.109 0.010–0.702 | 0.158±0.095 0.006–0.569 | 0.174±0.104 0.005–1.113 |
| Dendrite diameter (µm, mean ± std) | 0.60±0.10 | 0.56±0.06 | 0.57±0.09 | 0.89±0.24 | 0.82±0.20 | 0.86±0.25 | 1.31±0.14 | 1.32±0.16 | 1.38±0.19 |
| Dendrite length (µm, mean ± std) | 23.3±5.4 | 23.2±6.9 | 21.9±6.8 | 26.8±5.9 | 23.8±7.5 | 23.7±8.0 | 8.19±2.97 | 8.68±3.00 | 8.66±3.92 |
| Spine density (with/without synapse) by dendrite surface area (#/µm², mean ± std) | 1.72±0.30 | 2.17±0.33 | 2.00±0.34 | 0.76±0.22 | 0.78±0.23 | 0.84±0.23 | 1.83±0.30 | 2.16±0.44 | 1.97±0.51 |
| Process density (spines without synapse) by dendrite surface area (#/µm², mean ± std) | 0.04±0.03 | 0.06±0.06 | 0.07±0.08 | 0.12±0.07 | 0.10±0.06 | 0.15±0.09 | 0.08±0.06 | 0.13±0.07 | 0.19±0.10 |
| Synapse density (spines with synapse) by dendrite surface area (#/µm², mean ± std) | 1.69±0.30 | 2.12±0.34 | 1.93±0.32 | 0.64±0.20 | 0.68±0.22 | 0.69±0.19 | 1.76±0.28 | 2.04±0.41 | 1.78±0.46 |
| Nonperforated synapse density by dendrite surface area (#/µm², mean ± std) | 1.27±0.26 | 1.72±0.29 | 1.47±0.29 | – | – | – | – | – | – |
| Perforated synapse density by dendrite surface area (#/µm², mean ± std) | 0.41±0.13 | 0.36±0.10 | 0.44±0.18 | – | – | – | – | – | – |

*Table 2 continued on next page*

*Table 2 continued*

| | CA1 (stratum radiatum) | | | Cortex (M1+S1, layer 2) | | | Cerebellum | | |
|---|---|---|---|---|---|---|---|---|---|
| | W | EW | S | W | EW | S | W | EW | S |
| **Synapse density** (spines with synapse) by dendrite length (#/μm, mean ± std) | 3.14±0.60 | 3.69±0.50 | 3.41±0.50 | – | – | – | 7.86±1.39 | 9.07±1.73 | 8.29±1.82 |
| **Nonperforated synapse density** by dendrite length (#/μm, mean ± std) | 2.38±0.52 | 3.00±0.43 | 2.59±0.35 | – | – | – | – | – | – |
| **Perforated synapse density** by dendrite length (#/μm, mean ± std) | 0.77±0.23 | 0.63±0.16 | 0.78±0.31 | – | – | – | – | – | – |
| **Perforated synapses** % of synapses per mouse (mean ± std) | 24.5 ± 1.3% | 17.6 ± 1.9% | 23.2 ± 5.3% | – | – | – | – | – | – |
| **Oblique spines** % of synapses per mouse (mean ± std) | 4.8 ± 1.7% | 4.7 ± 0.9% | 5.2 ± 2.8% | 8.5 ± 2.2% | 10.8 ± 3.3% | 10.0 ± 1.7% | 7.0 ± 6.0% | 5.6 ± 3.5% | 1.8 ± 2.1% |
| **Incomplete synapses** (go off the image) % of spines with synapse per mouse (mean ± std) | 1.0 ± 0.9% | 1.7 ± 0.9% | 1.5 ± 1.5% | 2.2 ± 2.2% | 2.8 ± 5.2% | 4.0 ± 4.4% | 0.8 ± 1.5% | 1.2 ± 0.3% | 1.2 ± 0.7% |
| **Spines with spine apparatus** % of spines with synapse per mouse (mean ± std) | 8.2 ± 0.7% | 10.1 ± 3.1% | 13.5 ± 3.5% | 30.1 ± 4.0% | 29.2 ± 2.5% | 26.1 ± 4.5% | 2.2 ± 3.6% | 6.4 ± 9.8% | 1.8 ± 3.1% |
| **Spines with endosome/s** % of spines with synapse per mouse (mean ± std) | 34.8 ± 9.4% | 42.9 ± 12.4% | 41.3 ± 9.5% | 46.4 ± 11.1% | 67.2 ± 8.0% | 57.5 ± 10.0% | 96.3 ± 7.4% | 98.9 ± 1.5% | 99.8 ± 0.5% |
| **Synapses with coated vesicle** % of spines with synapse per mouse (mean ± std) | 2.8 ± 2.2% | 2.9 ± 1.3% | 1.8 ± 1.1% | 2.5 ± 1.8% | 4.3 ± 3.2% | 2.6 ± 1.7% | 1.5 ± 1.9% | 0.2 ± 0.2% | 0.1±0.2% |
| **Synapses with spinula** % of spines with synapse per mouse (mean ± std) | 19.4 ± 1.9% | 10.1 ± 3.8% | 14.0 ± 3.2% | 1.7 ± 2.0% | 1.3 ± 1.4% | 1.1 ± 0.5% | 1.6 ± 1.0% | 10.3 ± 13.5% | 22.7 ± 16.6% |

*Table 2 continued on next page*

*Table 2 continued*

| | CA1 (stratum radiatum) | | | Cortex (M1+S1, layer 2) | | | Cerebellum | | |
|---|---|---|---|---|---|---|---|---|---|
| | W | EW | S | W | EW | S | W | EW | S |
| **Synapses with MVB** % of spines with synapse per mouse (mean ± std) | 5.3 ± 2.3% | 2.7 ± 3.2% | 3.4 ± 3.2% | 6.4 ± 3.5% | 6.6 ± 2.9% | 3.3 ± 2.2% | 3.4 ± 4.1% | 5.7 ± 8.0% | 2.8 ± 3.9% |
| **Synapses with presynaptic mitochondrion** % of spines with synapse per mouse (mean ± std) | 30.3 ± 2.1% | 30.3 ± 3.7% | 31.6 ± 3.6% | 34.7 ± 4.2% | 35.1 ± 2.1% | 33.7 ± 3.3% | 58.0 ± 1.9% | 61.8 ± 7.5% | 57.3 ± 4.4% |
| **Branched synapses** % of spines with synapse per mouse (mean ± std) | 15.0 ± 3.7% | 15.8 ± 4.9% | 13.5 ± 1.6% | 13.9 ± 6.1% | 12.2 ± 3.2% | 13.7 ± 3.1% | 17.8 ± 3.3% | 21.0 ± 7.4% | 14.1 ± 2.4% |

**Table 3.** Parameter estimates for the linear mixed effect (LME) models used to assess changes in spine densities and proportions.

A square-root transformation was applied to each model. Residual analysis showed no evidence against normality or constant variance assumption.

Spine density (all spines) – sqrt(#/area)

| Random effects | Standard error | | |
|---|---|---|---|
| Mouse (intercept) | 0.0618 | | |
| Residual | 0.1426 | | |
| **Fixed effects** | **Level** | **Estimate** | **Standard error** |
| Intercept | | 1.4681 | 0.0426 |
| Condition | EW (reference) | 0 | 0 |
| | W | –0.1140 | 0.0590 |
| | S | –0.0757 | 0.0542 |

Spine density (spines with synapse) – sqrt(#/area)

| Random effects | Standard error | | |
|---|---|---|---|
| Mouse (intercept) | 0.0605 | | |
| Residual | 0.1349 | | |
| **Fixed effects** | **Level** | **Estimate** | **Standard error** |
| Intercept | | 1.4241 | 0.0412 |
| Condition | EW (reference) | 0 | 0 |
| | W | –0.0994 | 0.0569 |
| | S | –0.1002 | 0.0522 |

Spine density (branched synapses) – sqrt(#/area)

| Random effects | Standard error | | |
|---|---|---|---|
| Mouse (intercept) | 0.0350 | | |
| Residual | 0.1760 | | |
| **Fixed effects** | **Level** | **Estimate** | **Standard error** |
| Intercept | | 0.5920 | 0.0401 |
| Condition | EW (reference) | 0 | 0 |
| | W | –0.0934 | 0.0547 |
| | S | –0.2281 | 0.0503 |

Spine density (spines without synapse) – sqrt(#/area)

| Random effects | Standard error | | |
|---|---|---|---|
| Mouse (intercept) | 0.0387 | | |
| Residual | 0.1027 | | |
| **Fixed effects** | **Level** | **Estimate** | **Standard error** |
| Intercept | | 0.3443 | 0.0286 |
| Condition | EW (reference) | 0 | 0 |
| | W | –0.0791 | 0.0395 |
| | S | 0.0760 | 0.0363 |

Proportion of spines without synapse – sqrt(# without/# total)

*Table 3 continued on next page*

*Table 3 continued*

Spine density (all spines) – sqrt(#/area)

| Random effects | Standard error | | |
|---|---|---|---|
| Mouse (intercept) | 0.0270 | | |
| Residual | 0.0642 | | |
| Fixed effects | Level | Estimate | Standard error |
| Intercept | | 0.2331 | 0.0189 |
| Condition | EW (reference) | 0 | 0 |
| | W | −0.0386 | 0.0240 |
| | S | 0.0674 | 0.0262 |

Proportion of synapses with spinula – sqrt(# with spinula/# total)

| Random effects | Standard error | | |
|---|---|---|---|
| Mouse (intercept) | 0.1285 | | |
| Residual | 0.0898 | | |
| Fixed effects | Level | Estimate | Standard error |
| Intercept | | 0.4276 | 0.0669 |
| Condition | EW (reference) | 0 | 0 |
| | W | −0.1730 | 0.0944 |
| | S | 0.0733 | 0.0862 |

## Branched spines and branched synapses

How can we account for the sleep/wake difference in the proportion of spines with and without a synapse? To address this question we focused on the branched synapses, which are defined as two (or sometimes more) distinct synapses housed in two (or sometimes more) spines that share the same neck, called branched spines (*Figure 4A*). In the cerebellum, like in the cerebral cortex and in the CA1 stratum radiatum, most synapses are harbored in single spines but in all three regions a sizable fraction of synapses and spines, around 15%, are branched. While the density of single synapses was similar across the three groups (mean ± std; W=1.47 ± 0.25; EW = 1.66 ± 0.34; S=1.62 ± 0.41), a significant effect of condition was found for the density of branched synapses (p=0.0013). The absolute density of branched synapses was twofold higher in both wake groups relative to the sleep group (mean ± std; W=0.29 ± 0.18; EW = 0.38 ± 0.21; S=0.16 ± 0.13), and post hoc tests confirmed that these differences were significant (S<EW, p=2e-5; S<SW, p=0.0141; W=EW, p=0.2010) (*Figure 4B*). The absolute density of branched spines, which almost always harbor synapses, also showed a significant effect of condition (p=0.0179), although post hoc tests found a significant difference only between S and EW mice (*Figure 4C*). In relative terms, branched synapses accounted for 15.1 ± 3.6% of all spines in W mice, 16.9 ± 6.3% in EW mice, and 7.6 ± 1.2% of all spines in S mice (mean ± SD per mouse; *Figure 4D*). In almost all cases (93%) the branched synapses contacted boutons belonging to two different parallel fibers (*Figure 4E*); in rare cases there were three branched synapses, each contacting a different fiber (*Figure 4F*). In only 7% of cases the branched synapses contacted two boutons of the same fiber (*Figure 4G*) or the same bouton (*Figure 4H*). For almost all branched synapses it was possible to track the axon far enough in the stack to see that the same parallel fiber established another synapse with a different dendrite (~2/3 cases) or the same dendrite (~1/3 cases).

In sum, we found that the absolute number of branched synapses is twice higher after wake (both spontaneous and extended) than after sleep, on average by ~0.2 synapses per dendrite surface area. Correspondingly, the proportion of branched synapses over all spines decreases from an aggregate of ~16% in the wake conditions to ~8% after sleep. Conversely, the absolute number of spines (unbranched or branched) lacking a synapse ('naked' spines) is higher after sleep, on average by ~0.1 spines per dendrite surface area and, correspondingly, the proportion of naked spines is ~10% after

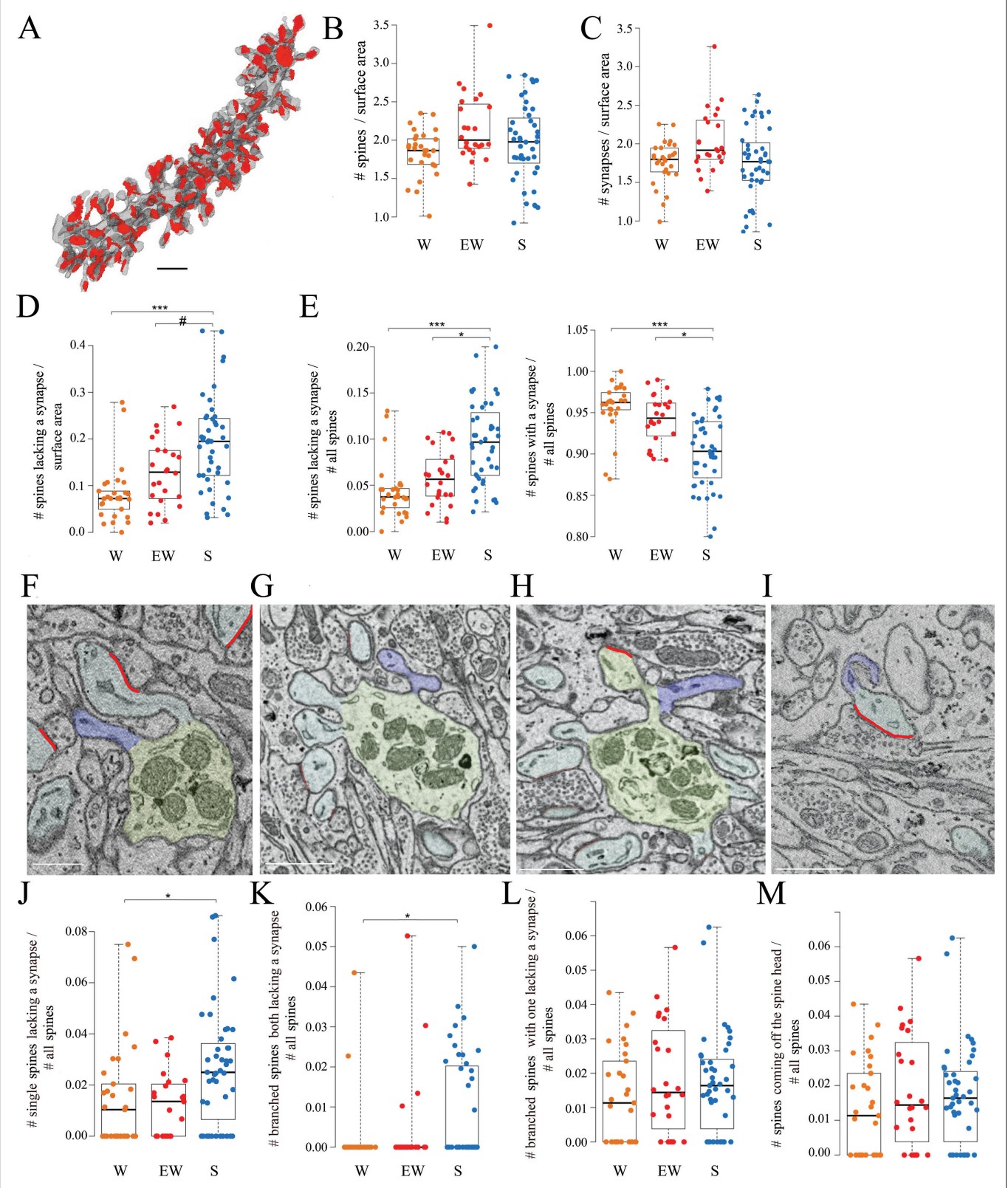

**Figure 3.** Spines and synapses across groups. (**A**) 3D reconstruction of a dendritic branch (spines and dendrite in gray, axon-spine interfaces [ASIs] in red). (**B, C**) Absolute density of spines and synapses (N per dendrite surface area in µm²) in each experimental group. Each dot is one dendrite. (**D**) Absolute density of spines without a synapse (N per dendrite surface area in µm²) in each experimental group. Each dot is one dendrite. (**E**) Relative density of spines without and with a synapse in each experimental group. Each dot is one dendrite. (**F–I**) 2D images showing examples of spines without

*Figure 3 continued on next page*

*Figure 3 continued*

a synapse, including a single spine (F, in blue), branched spines both lacking a synapse (G, in blue), branched spines with one lacking a synapse (H, in blue), and spines without a synapse coming off the head of a spine with a synapse (I, in blue). (**J–M**) Relative density of the four types of spines lacking a synapse. Each dot is a dendrite. Scale bars = 1 μm *, p<0.05; ***, p<0.001; #, p=0.0906. Source data are provided as a source data file (dendrite data. csv).

sleep compared to ~5% after wake. Altogether, the decrease in branched synapses after sleep can account for the overall increase in the proportion of naked spines after sleep (taking into account the variability across dendrites). These results suggest that, in the course of sleep, branched spines lose one synapse, or both synapses, or are converted into single spines after losing both the synapse and the housing spine, while the opposite changes occur in the course of wake.

## Spine morphology and ASI

The spines were highly homogeneous in shape, mostly club-shaped, and their synapses were usually located on the side of the spine head rather than on its tip (*Figure 5A*), in agreement with previous reports (*Spacek and Hartmann, 1983*; *Harris and Stevens, 1988*; *Napper and Harvey, 1988a*). Most spines contained tubules and vesicles of the smooth endoplasmic reticulum (SER), as previously described (*Harris and Stevens, 1988*; *Martone et al., 1993*), while only a few spines housed elements of the non-SER such as coated vesicles and multivesicular bodies (*Table 2*). The ASI size did not vary greatly within or across dendritic branches (range from 0.0053 to 1.1131 μm$^2$). In the same mice, we previously found that ASI sizes follow a log-normal distribution in the cerebral cortex (*de Vivo et al., 2017*) and a bimodal distribution in the CA1 stratum radiatum of the hippocampus (*Spano et al., 2019*). In the cerebellum, the distribution of ASI sizes was unimodal, with mainly small/medium ASIs, but not enough very large ASIs to be considered a log-normal distribution (*Figure 5B*). Fitting an LME model for ASI (square-root transformation) with condition and dendrite diameter as fixed effects and mouse and dendrite as random effects revealed no effect of condition, neither on the mean ASI (p=0.0664; post hoc tests: S=EW p=0.2182; S=W p=0.4861; SW <EW p=0.0279) (*Figure 5C*) nor on the cumulative ASI (sum of ASIs per dendrite area; p=0.9098) (*Table 4*). The average ASI value was significantly lower in branched synapses compared to non-branched synapses (p=0.0002).

Other features that were measured inside each spine, including the presence of mitochondria and of components of the endoplasmic reticulum, did not change across experimental groups (*Table 2*). The only exception was the proportion of synapses with a spinula, which was higher in S mice than in W mice but highly variable in EW mice (effect of condition, p=0.0386; post hoc tests, S>W, p=0.0115; S>EW, p=0.6708) (*Table 2*).

## Discussion

Sleep and wake profoundly affect synaptic activity and plasticity in the cerebral cortex and the hippocampus, with electrophysiological, molecular, and ultrastructural evidence pointing to overall excitatory synaptic strength being higher after wake and lower after sleep (*Tononi and Cirelli, 2014*; *Cirelli and Tononi, 2020*; *Cirelli and Tononi, 2021*). The cerebellum contains the largest number of neurons and excitatory synapses in the brain – most of them between parallel fibers and Purkinje cells – and is known to be highly plastic. However, it was hitherto completely unknown whether sleep and wake can affect cerebellar synapses, and specifically the number and strength of the parallel fiber synapses. Here, we addressed this question by applying serial electron microscopy to obtain the full 3D reconstruction of more than ~7000 of spines located on the distal dendrites of Purkinje cells.

While the vast majority of spines carry a synapse, we found that a sizable minority do not. Remarkably, these 'naked' spines account for only ~5% of all spines after spontaneous or forced wake (~0.1 spines per dendrite surface area) but for ~10% of all spines after sleep (~0.2 spines per dendrite surface area). Most spines contacted by a parallel fiber carry a single synapse (*Napper and Harvey, 1988a*; *Napper and Harvey, 1988b*), but a fraction of spines share the same neck and carry two or occasionally three synapses (branched synapses on branched spines). By investigating branched synapses, we then found that they also decreased from wake to sleep, and that the reduction in the absolute number of branched synapses can account for the overall change in the proportion of naked spines. Specifically, there were on average ~0.3 branched synapses per dendrite surface area after

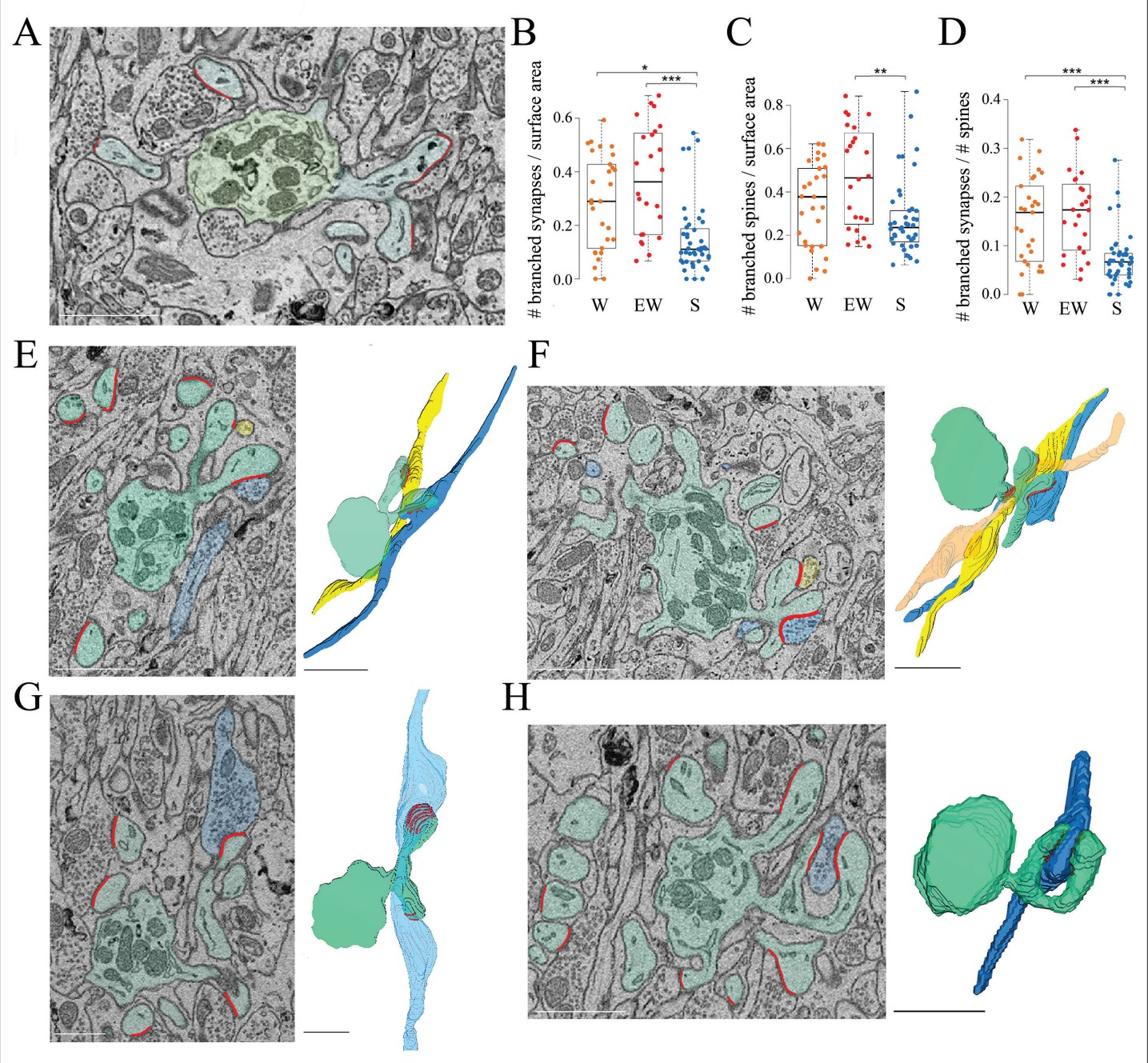

**Figure 4.** Branched spines and synapses across groups. (**A**) 2D image showing an example of two branched synapses, that is, housed in two spines that share the same spine neck, called branched spines (synapses in blue, axon-spine interfaces [ASIs] in red). The dendritic shaft (green) contains multiple mitochondria. (**B**) Absolute density of branched synapses in each experimental group. Each dot is one dendrite. (**C**) Absolute density of branched spines in each experimental group. Each dot is one dendrite. (**D**) Relative density of branched synapses in each experimental group. Each dot is one dendrite. (**E, F**) Examples of branched synapses contacting two (in E) or three (in F) different parallel fibers. Raw image (left) and 3D reconstruction (right). (**G**) Example of branched synapses contacting two boutons of the same fiber. (**H**) Example of branched synapses contacting the same bouton. Scale bars = 1 μm. *, p<0.05; **, p<0.005; ***, p<0.001. Source data are provided as a source data file (dendrite data.csv).

sleep deprivation (~17% of all spines), but only ~0.1 after sleep (~8% of all spines). Thus, in the course of sleep branched spines may lose one synapse or both synapses, or they may be converted to single spines after losing both the synapse and the housing spine, whereas the opposite changes occur in the course of wake. These changes occur with no significant changes in the total number of spines

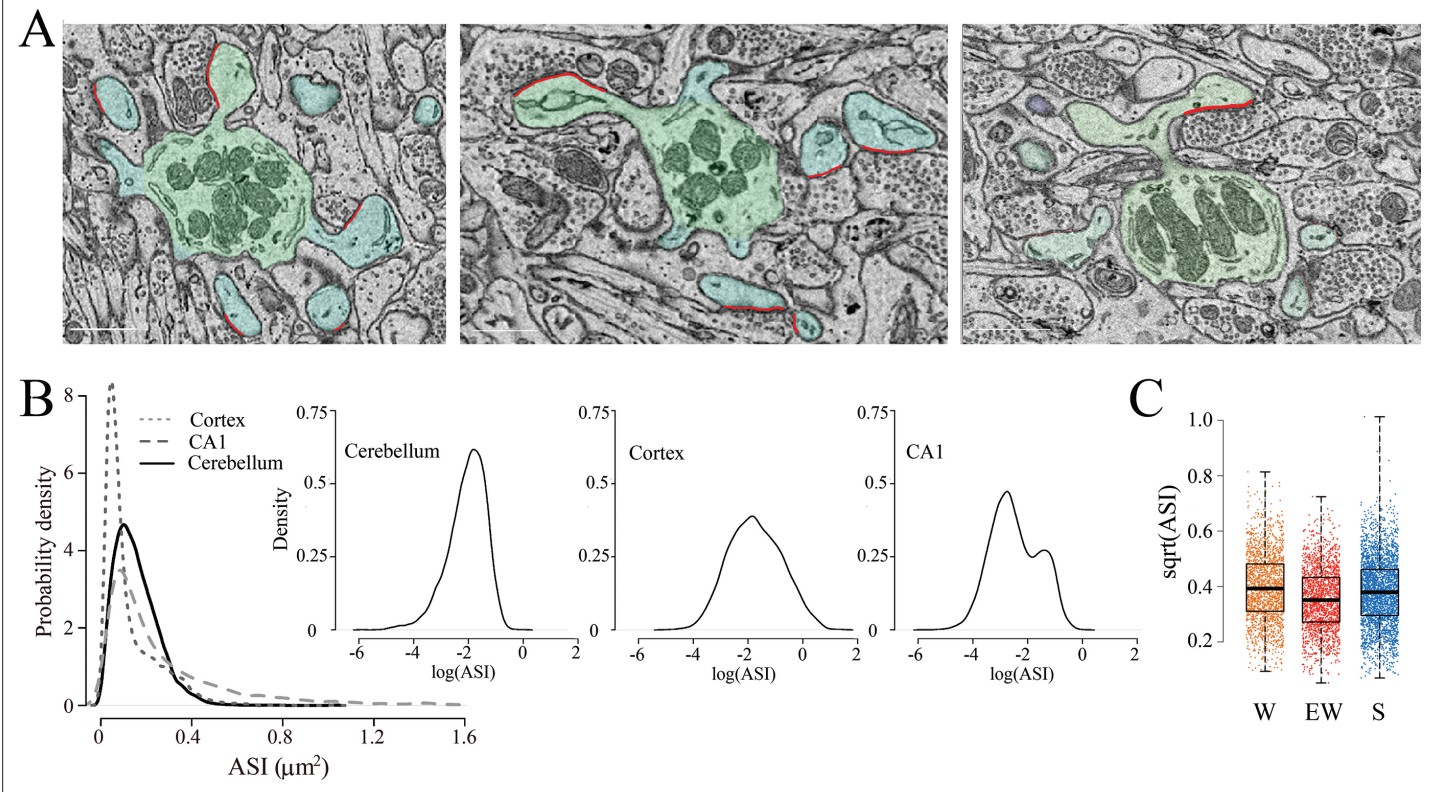

**Figure 5.** The size of the axon-spine interfaces (ASI) across groups. (**A**) Examples of spines with a synapse (ASI in red). Scale bars = 1 μm. (**B**) Distribution of ASI sizes in cerebellum, cerebral cortex (replotted from *de Vivo et al., 2017*), and hippocampus (CA1, replotted from *Spano et al., 2019*) across all three experimental groups. Insets, same on a log scale. (**C**) Distribution of ASI sizes (sqrt, square-root transformation) in each group. Each dot is a synapse. Source data are provided as a source data file (synapse data.csv).

and parallel fiber synapses with behavioral state, presumably because naked spines and branched synapses constitute a minority of all spines and synapses.

We also found that sleep and wake do not affect the size of the ASI, a structural measure of synaptic strength (*Desmond and Levy, 1988*; *Buchs and Muller, 1996*; *Cheetham et al., 2014*). This is in contrast with our findings in cortex and hippocampus, where sleep and wake lead to changes in the average strength of the synapses rather than in their number (*Cirelli and Tononi, 2021*). Thus, the effects of sleep and wake point in the same direction in all three brain regions that we have examined so far, with more and/or stronger synapses after wake than after sleep. However, the underlying mechanisms are quite distinct. In the cerebellum sleep and wake affect the number of branched synapses and the relative number of spines that carry a synapse but not the average synaptic strength. In cerebral cortex and hippocampus instead, sleep and wake mainly affect the size of the ASI but not the number of synapses. These regional differences are notable, since the current study utilized the same 14 mice previously used for the analysis of CA1 (*Spano et al., 2019*) and 8 of them were also used for the analysis of the primary motor and sensory cortex (*de Vivo et al., 2017*), but they are not unexpected. Specifically, the lack of changes in ASI size is less surprising if one considers the striking difference in the distribution of synaptic sizes between cortex and hippocampus on one hand, and the cerebellum on the other hand. In the first two regions the distributions are log-normal and bimodal, respectively, meaning that synaptic sizes (and strengths) cover a wide range and include large synapses. By contrast, the sizes of the parallel fiber synapses follow a unimodal distribution, meaning that these synapses tend to be all of the same small-medium size. This narrow range of sizes strongly suggests that, in general, plastic changes in these synapses are implemented more via all or none changes (adding or removing a synapse) than through graded changes in size.

**Table 4.** Parameter estimates for the linear mixed effect (LME) models used to assess changes in axon-spine interfaces (ASI). A square-root transformation was applied to the ASI values to ensure the residuals had an approximate Gaussian distribution. Residual analysis showed only minor departures from normality and constant variance in the ASI models, and no evidence against the assumptions in the cumulative ASI model.

ASI − sqrt(ASI)

| Random effects | Standard error | | |
|---|---|---|---|
| Dendrite (intercept) | 0.0134 | | |
| Mouse (intercept) | 0.0164 | | |
| Residual | 0.1219 | | |

| Fixed effects | Level | Estimate | Standard error |
|---|---|---|---|
| Intercept | | 0.3797 | 0.0227 |
| Dendrite diameter | Continuous (linear) | −1e-7 | 2e-5 |
| Condition | EW (reference) | 0 | 0 |
| | W | 0.0331 | 0.0129 |
| | S | 0.0197 | 0.0118 |

ASI (with branched) − sqrt(ASI)

| Random effects | Standard error | | |
|---|---|---|---|
| Dendrite (intercept) | 0.0132 | | |
| Mouse (intercept) | 0.0168 | | |
| Residual | 0.1217 | | |

| Fixed effects | Level | Estimate | Standard error |
|---|---|---|---|
| Intercept | | 0.3812 | 0.0227 |
| Dendrite diameter | Continuous (linear) | 7e-7 | 2e-5 |
| Condition | EW (reference) | 0 | 0 |
| | W | 0.0333 | 0.0132 |
| | S | 0.0184 | 0.0120 |
| Branched | No (reference) | 0 | 0 |
| | Yes | −0.0217 | 0.0052 |

ASI density (sum ASI/surface area)

| Random effects | Standard error | | |
|---|---|---|---|
| Mouse (intercept) | 0.0425 | | |
| Residual | 0.0608 | | |

| Fixed effects | Level | Estimate | Standard error |
|---|---|---|---|
| Intercept | | 0.3159 | 0.0247 |
| Condition | EW (reference) | 0 | 0 |
| | W | −0.0042 | 0.0349 |
| | S | −0.0131 | 0.0316 |

## Branched synapses as eloquent synapses?

What could be the significance of these marked changes in the number of branched synapses between wake and sleep? An intriguing possibility is that in the cerebellum branched synapses – two or three in spines sharing the same neck – may be 'eloquent', whereas non-branched synapses – one per spine – may be silent or only partially functional. The notion that a majority of parallel fiber synapses may be

silent was first suggested by the finding that the receptive field of Purkinje cells is much smaller than the one of the granule cells thought to project to them (*Ekerot and Jörntell, 2001*). Experiments in slices have determined that at least 80% of the parallel fiber synapses are either silent or too weak to produce a detectable somatic response (*Isope and Barbour, 2002*). This conclusion was reached because only 7% of stimulated granule cells generated electric responses in the soma of Purkinje cells, as opposed to the expected ~54% based on the anatomical count of spines and synapses (*Napper and Harvey, 1988a*; *Napper and Harvey, 1988b*). A recent study in slices also found that more than 70% of parallel fiber synapses did not produce detectable fast postsynaptic responses (*Ho et al., 2021*). This proportion was the same in young and adolescent mice, suggesting that it is a stable feature of the cerebellum even during development (*Ho et al., 2021*). In short, there is evidence that perhaps only around 10–20% of parallel fiber synapses may be eloquent, that is, capable of eliciting currents in the soma of Purkinje cells. This proportion, obtained with functional tests, is similar to the proportion of branched synapses as determined morphologically in this study (~17% after extended wake and ~8% after sleep). A model of cerebellar networks also suggests that only coincident inputs from the granule cells sum up effectively and drive a Purkinje cell to fire (*Brunel et al., 2004*). It may be that the simultaneous activation of branched synapses may be ideally suited to promoting synaptic signal transfer to the soma, as also suggested by modeling studies (*Rusakov et al., 1996*).

## Branched synapses as coincidence detectors for associative plasticity?

Modeling studies also suggest that, given the extraordinary number of parallel fibers synapses contacting each Purkinje cell – ~175,000 for each Purkinje cell in the rat – (*Napper and Harvey, 1988a*), information storage may be optimized under conditions in which only a small proportion of synapses is strong enough to be 'eloquent' (*Brunel et al., 2004*). If structurally branched synapses correspond to functionally eloquent ones, another intriguing possibility is that branched synapses may also represent the structural counterpart of learning through coincidence detection in the activity of afferent fibers during waking behaviors. As also shown here, the two synapses on a branched spine are typically (93% of cases) contributed by different parallel fibers. It has been estimated that almost 50% of the parallel fibers traversing the Purkinje cells do not synapse with them (*Napper and Harvey, 1988a*). Thus, parallel fibers may have the potential to establish many new contacts with their target cells. Coincidences in firing are thought to mediate associative plasticity through various synaptic mechanisms (*Hashmi et al., 2013*). Therefore, the increase in the number of branched synapses we observed after periods of wakefulness may represent a structural correlate of at least some forms of learning in the cerebellum. In other brain structures, such as the dentate gyrus of the hippocampus, the number of branched synapses increases after high-frequency stimulation, kindling, and long-term synaptic potentiation (*Geinisman et al., 1989*; *Trommald et al., 1996*; *Andersen and Soleng, 1998*; *Dhanrajan et al., 2004*).

## Sleep-dependent decrease in branched spines as elimination of spurious coincidences?

Given the large number of parallel fiber synapses on each Purkinje cell and the spontaneous activity of the massive number of granule cells from which they originate, it is inevitable that coincident firing may occur not just as a result of adaptive behaviors, but also spuriously, either through trial and error or simply because of chance. In such a scenario, there must be ways to consolidate adaptive memories and weaken or erase maladaptive or spurious ones (*Tononi and Cirelli, 2014*). Periods of off-line activity during sleep, in which brain circuits are systematically reactivated in a changed neuromodulatory milieu, are ideally suited to such synaptic down-selection (*Tononi and Cirelli, 2014*).

In the cortex and hippocampus, while branched synapses also represent 12–16% of all synapses, their number does not change with sleep and wake (*Table 2*). Instead, unlike in the cerebellum, sleep is associated with net synaptic down-selection primarily through the removal of AMPA receptors from the spine head (*Cirelli and Tononi, 2021*). There are several reasons why, in the case of cerebellar parallel fibers, down-selection of branched synapses may constitute the primary mechanism of synaptic renormalization during sleep. If the hypothesis that branched synapses correspond to eloquent synapses is correct, then their pruning would represent the natural way to erase the structural traces of spurious coincidences. Moreover, as shown here, parallel fiber synapses, compared to forebrain excitatory synapses, are small and homogeneous in size, which is not surprising given

their immense number. This may leave little room for decreasing their strength through the removal of AMPA receptors. Small spines with fewer AMPA receptors may be more susceptible to disappear after several hours of sleep because they lack the strong cytoskeleton network that usually anchor the receptors to the membrane. Branched synapses may be especially vulnerable because we found that their average strength, as measured by the size of the ASI, is even smaller than that of non-branched synapses. In turn, the removal of branched synapses during sleep may trigger the disappearance of the postsynaptic density (PSD) followed by the detachment of the presynapse, hence the disappearance of the synapse and, in some cases, even of the spine housing that synapse. This hypothesis could be tested directly in vivo using two-photon repeated imaging to track the same cerebellar synapses across the sleep/wake cycle, as we recently did for cortical synapses (*Miyamoto et al., 2021*).

It should also be mentioned that the glutamatergic synapses of the cerebellum follow rules of plasticity that partly diverge from those prevalent in the forebrain. A recent study *Gutierrez-Castellanos et al., 2017* found that the long-term potentiation of the parallel fiber synapses was required for vestibulo-cerebellar motor learning and relied on AMPA receptors containing the GluR3 subunit, rather than the GluR1 subunit commonly involved in cortical and hippocampal plasticity (*Diering and Huganir, 2018*). GluR3-mediated synaptic potentiation did not involve the trafficking of the receptors but resulted from an increase in open channel probability (*Gutierrez-Castellanos et al., 2017*). Thus, at least some forms of synaptic potentiation in the parallel fiber synapses do not rely on the insertion of AMPA receptors in the synaptic membrane and are not expected to increase the size of the synapses and the size of the ASI. On the other hand, the endocytosis of GluR2-containing AMPA receptors is key for the long-term depression of the parallel fiber synapses (*Chung et al., 2003*; *Steinberg et al., 2006*) as it is the case for many cortical and hippocampal synapses (*Diering and Huganir, 2018*). Of note, in the adult cerebellum the glutamate receptor delta 2 is specifically expressed in the parallel fiber synapses and has been involved in long-term depression and in the maintenance of these synapses (*Mandolesi et al., 2009*). In a genome-wide transcriptomic analysis of cerebellar transcripts affected by sleep and wake, we found that the mRNA coding for the delta 2 subunit was upregulated after sleep (*Cirelli et al., 2004*). Whether changes in the expression of this receptor contribute to the synaptic changes described here is not known and could be explored in future studies.

## Limitations and conclusions

Several studies have shown that the cerebellar cortex is involved in the early phases of cerebellar learning, while later stages involve the cerebellar and vestibular nuclei downstream (*D'Angelo et al., 2016*; *Canto et al., 2017*). Because of the labor-intensive nature of ultrastructural studies, we could only examine structural changes in synapses across sleep and wake in a restricted portion of cerebellar cortex that contains a relatively homogeneous population of Purkinje cells. In the area that we targeted, the vermal lobule VI, most or all Purkinje cells express the molecular marker zebrin II (*Cerminara et al., 2015*). Zebrin positive and zebrin negative Purkinje cells differ in many structural and functional parameters, from the dominant type of climbing fiber input that they receive to the level of intrinsic excitability and spiking activity (*Zhou et al., 2014*; *Cerminara et al., 2015*; *Viet et al., 2022*). The extent to which the current results extend to zebrin negative Purkinje cells is unknown. On the other hand, zebrin negative cells express stronger intrinsic and synaptic plasticity relative to zebrin positive cells (*Wadiche and Jahr, 2005*; *Viet et al., 2022*).

In the cerebral cortex, we previously reported that the sleep/wake changes in ASI size are accompanied by several changes in the peripheral astrocytic processes surrounding many cortical synapses. Specifically, these processes contain more numerous but smaller glycogen granules after spontaneous and forced wake compared to sleep, suggesting increased glycogen turnover to meet the high energy demand of wake (*Bellesi et al., 2018*). Moreover, after a few hours of sleep deprivation astrocytic processes get closer to the synaptic cleft, likely reflecting an increased need for glutamate clearance (*Bellesi et al., 2015*), and astrocytic phagocytosis of synaptic elements is increased (*Bellesi et al., 2017*). Bergmann glia, the astrocytic cells of the cerebellum, establish extensive contacts with parallel fiber synapses and their distal processes respond to the electrical stimulation of the parallel fibers with localized increases in calcium levels (*Grosche et al., 1999*). Calcium signaling in Bergmann glia can, in turn, modulate the activity of neighboring Purkinje cells (*Wang et al., 2012*), in line with strong neuron-glia crosstalk. Moreover, contrary to cortical astrocytes, Bergmann glia express AMPA receptors, which are required for the normal development of the parallel fiber synapses (*Saab et al., 2012*).

Bergmann glia is also routinely involved in the phagocytosis of synaptic and extrasynaptic elements, a process that increases after motor learning (*Morizawa et al., 2022*). Thus, given the multiple ways Bergmann glia can modulate cerebellar activity (*De Zeeuw and Hoogland, 2015*), and specifically the parallel fiber synapses that are the focus of the current study, future experiments should test whether sleep and wake affect the structure and function of these cells.

Very little is known about the effects of sleep and wake on cerebellar physiology (*Canto et al., 2017*), and the available evidence is mostly confined to the Purkinje cells. Both their simple spike activity, which is driven by the excitatory inputs from parallel fiber synapses, and their complex spike activity, which reflects the excitatory input from a single climbing fiber, are lower during non-rapid eye movement (NREM) sleep than during wake or REM sleep (*Mano, 1970*; *Marchesi and Strata, 1971*; *Hobson and McCarley, 1972*; *Harlay et al., 1974*; *Canto et al., 2017*). The response of Purkinje cells to microiontophoretically applied glutamate is also smaller in NREM sleep than in wake (*Andre and Arrighi, 2001*). This effect is not accounted for by changes in spontaneous firing rate but may be associated with the decreased level of acetylcholine and noradrenaline in NREM sleep, which is observed across the brain, including in the cerebellum (*Andre and Arrighi, 2001*). However, whether the sleep-wake cycle is also accompanied by changes in the number or strength of synaptic connections was completely unknown. As shown here through serial electron microscopy, the cerebellum, like the cerebral cortex and the hippocampus, also shows ultrastructural synaptic changes consistent with an overall upregulation during wake and down-selection during sleep (*Tononi and Cirelli, 2014*). The selective changes in branched synapses across wake and sleep also suggest that, during wake, coincidences of firing over parallel fibers may translate into the formation of synapses coincident over the same branched spine, which may be especially effective at driving the soma of Purkinje cells. During sleep, off-line activity over parallel fibers may instead lead to the pruning of branched synapses that were formed due to spurious coincidences (*Figure 6*). While speculative, this interpretation is consistent with electrophysiological, anatomical, and modeling studies showing that (1) many parallel fiber synapses are silent and (2) given the exorbitant number of these synapses, information storage is promoted when only some of them are effective (eloquent).

## Materials and methods
### Mice

Homozygous B6.Cg-Tg(Thy1-YFP)16Jrs/J transgenic mice (IMSR Cat# JAX:003709, 78RRID:IMSR_ JAX:003709) expressing yellow fluorescent protein (YFP) in a subset of cortical pyramidal neurons (*Feng et al., 2000*) were used, as in our previous ultrastructural studies in the cerebral cortex and hippocampal CA1 region (*de Vivo et al., 2017*; *Spano et al., 2019*). Both female and male mice were used, balanced in number within each of the three experimental groups. All mice were around 1 month of age at the time of the experiment. In this mouse strain, the sleep/waking pattern and sleep homeostatic regulation at this age are very similar to those of adult mice (*Nelson et al., 2013*; *Cirelli and Tononi, 2020*). All animal procedures followed the National Institutes of Health Guide for the Care and Use of Laboratory Animals, and facilities were reviewed and approved by the IACUC of the University of Wisconsin-Madison and were inspected and accredited by AAALAC (animal protocol M005697).

### Experimental design
Three groups of mice were used, including six sleeping mice (S, three females), four sleep deprived mice (extended wake, EW, two females), and four spontaneously awake mice (W, two females). Sample size was determined based on past experience and pilot experiments. Our previous study describing sleep/wake changes in the CA1 region. *Spano et al., 2019*, used the same 14 mice, and 8 of them were also used for the analysis of the cerebral cortex (*de Vivo et al., 2017*). The brain of S mice was collected during the light period (3.30–5.30 pm) at the end of a long period of sleep (>45 min, interrupted by periods of wake lasting less than 4 min), and after spending at least 75% of the previous 6–8 hr asleep. Brain collection for the EW mice occurred at the same time as for the S mice, but they were exposed to novel objects and other stimuli (e.g., tapping of the cage) to keep them awake during the first 6–8 hr of the day. The method of sleep deprivation reduces sleep by more than 95%

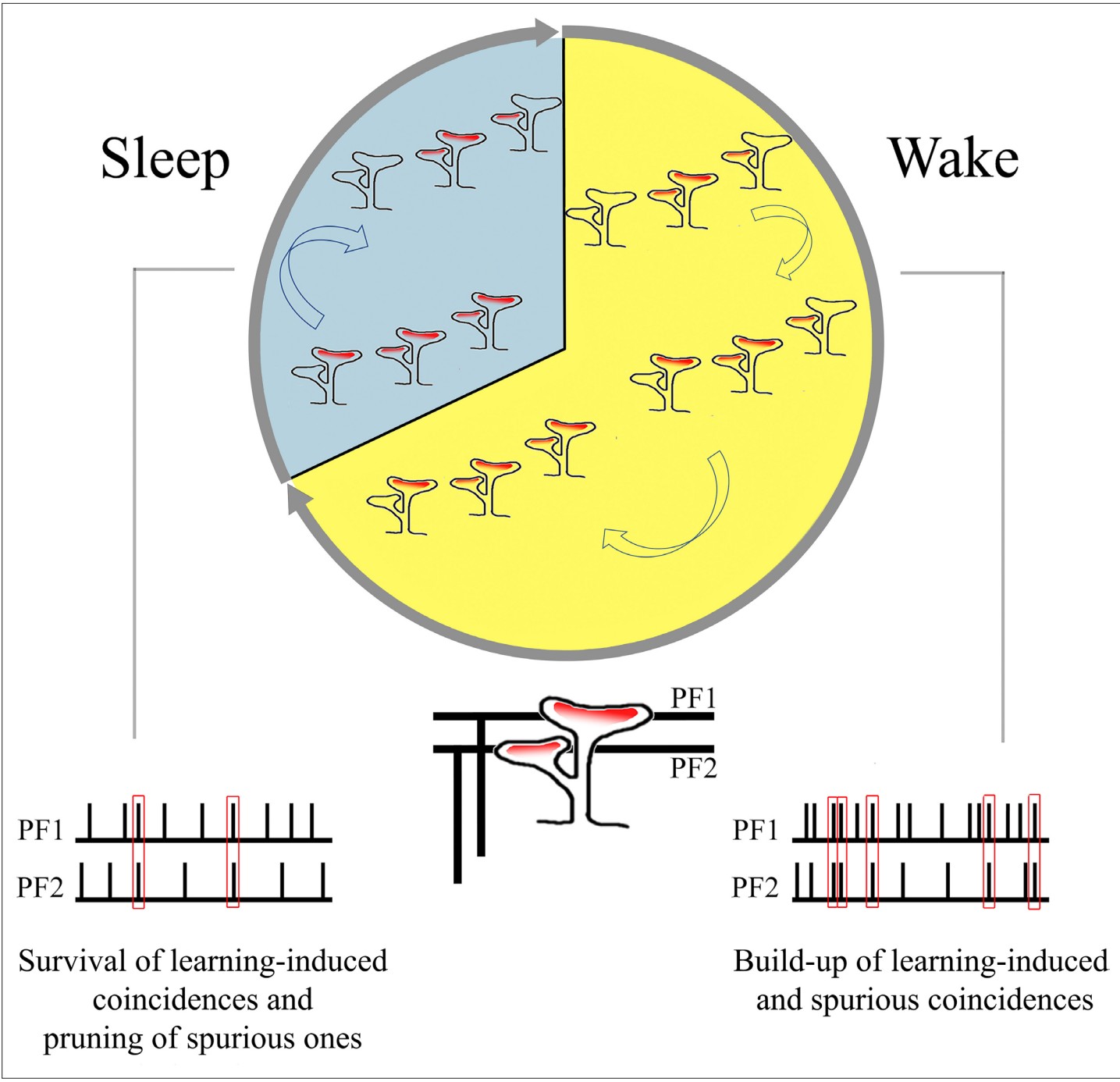

Survival of learning-induced
coincidences and
pruning of spurious ones

Build-up of learning-induced
and spurious coincidences

**Figure 6.** Summary of the main findings: increase in branched synapses in wake compared to sleep and its putative functional consequences.

(*Bellesi et al., 2013*; *Bellesi et al., 2015*). The brain of W mice was collected during the dark phase (~2–3:30 am) at the end of a long period of wake (>1 hr, interrupted by periods of sleep of <5 min), and after spending at least 70% of the previous 6–7 hr awake. Note that by using three groups of mice, we ensure that the differences between S and EW mice cannot be accounted for by circadian factors. However, additional experiments under constant light and dark conditions, or in clock gene mutant mice, would be necessary to rule out clock effects when comparing S and W mice.

Mice were housed in environmentally controlled recording chambers with free access to food and water (12 hr:12 hr light-dark cycle; lights on at 8 am). At night mice had access to a running wheel and one to two novel objects were introduced in their cage for enrichment and to promote the light/

dark entrainment of the rest/activity cycle. Running wheels and novel objects were removed during the light period, except in the case of EW mice. To avoid tissue damage mice were not implanted with electrodes for polysomnographic recording, and their sleep/wake behavior was monitored continuously using infrared cameras (OptiView Technologies). This method consistently estimates total sleep time with an accuracy exceeding 90% (*Maret et al., 2011*). Motor activity was quantified by custom-made video-based motion detection algorithms (MATLAB; see details in *Bellesi et al., 2013*).

## Staining for electron microscopy

Under deep isoflurane anesthesia (3% volume to oxygen) mice were transcardially perfused with normal saline (0.9%, 41°C, 30 s) followed by 2.5% glutaraldehyde and 4% paraformaldehyde dissolved in 0.1 M sodium cacodylate buffer (pH 7.4, 41°C, 10 min). After removal brains were kept in the same fixative overnight at 4°C, and then sliced on a vibratome. The slices (120 µm in thickness) were kept in a cryoprotectant solution until the day of processing. Small blocks of tissue (1 mm$^2$) from the vermal lobule VI region (from bregma, AP –6.8 mm, lateral 0.2 mm) were excised under a stereomicroscope and stained (details in *Wilke et al., 2013*; *de Vivo et al., 2017*). Staining was performed blind to experimental condition. Briefly, after several rinses in cacodylate buffer, the tissue was (1) incubated with a solution of 1% potassium ferrocyanide/2% osmium tetroxide (in the dark for 1 hr on ice); (2) exposed to a solution of 1% thiocarbohydrazide (20 min at room temperature); (3) placed in 2% osmium tetroxide (30 min); (4) incubated with 1% uranyl acetate (2 hr at room temperature followed by overnight at 4°C). The next day, the tissue was (1) stained with a solution of lead aspartate (30 min at 60°C, pH 5.5); (2) dehydrated using ice-cold solutions of freshly prepared 35%, 50%, 75%, 80%, 90%, 95%, and 100% ethanol; (3) placed in propylene oxide (2×10 min); (4) impregnated with 25%, 50%, 75% Durcupan ACM resin (Electron Microscopy Science) mixed with propylene oxide (2 hr each). Finally, the tissue was placed in fresh 100% Durcupan several times, flat embedded with ACLAR embedding film (Electron Microscopy Science) and kept in an oven at 60°C for 48–72 hr. After polymerization, the stained tissue was excised under a stereomicroscope and attached on the tip of a metal pin using conductive epoxy resin (Chemtronics) to minimize charging during imaging.

## Imaging

Samples were imaged in a $\Sigma$IGMA VP field emission scanning electron microscope with the following parameters: aperture 30 µm; high vacuum, acceleration voltage 1.7 kV, image size of 5000 by 5000 pixels, image resolution (xy plane) of 4 nm. One or two stacks of ~500 images each were acquired per animal (~20 × 20 × 25 µm$^3$) in the posterior vermis (lobule VI). Images were Gaussian filtered and automatically aligned using the open-source software Fiji (*Schindelin et al., 2012*). Ultrathin sections were cut at a nominal thickness of 50 nm. For each stack, the mean actual section thickness was estimated using the cylindrical diameters method (*Fiala and Harris, 2001*) and was similar across the three groups (mean ± std in nm; S=51.02 ± 1.67; W=52.18 ± 2.1; EW = 50.30 ± 3.5). Dendritic segments and all their spines were segmented manually in TrakEM2 (*Cardona et al., 2012*) by five trained annotators who were blind to experimental condition. We randomly selected spiny dendritic segments whose length was at least 3.122 µm (mean ± std in µm; S=8.66 ± 3.92; W=8.19 ± 2.97; EW = 8.68 ± 3.00; LME model p=0.8635) and whose diameter ranged between 0.940 and 1.966 µm. Distribution of dendritic diameters was balanced across experimental groups (mean ± std in µm; S=1.40 ± 0.19; W=1.31 ± 0.14; EW = 1.32 ± 0.16; LME model p=0.3579). We targeted the superficial part of the molecular layer, around 20–30 µm from the pial surface, where the distal dendritic branchlets house mostly, if not exclusively, parallel fiber synapses (*Ichikawa et al., 2016*).

All protrusions were defined as 'spines' (as suggested in *Holtmaat and Svoboda, 2009*), including spines with synapses and spines lacking synapses. A synapse was defined by the presence of a presynaptic bouton with at least two synaptic vesicles within a 50 nm distance from the cellular membrane facing the spine, a visible synaptic cleft and PSD. Branched synapses (housed in two or more spines sharing the same spine neck) were counted as two (or more) synapses. In total, 96 dendritic branches were segmented (S=43; W=29; EW = 24). All segmentation data were tested for accuracy and consistency by the same experienced tracers (SSL, CC).

As in previous studies (*de Vivo et al., 2017*; *Spano et al., 2019*) the ASI was traced at the interface between the spine head and the presynaptic terminal or bouton, and computed as described in *Bellesi et al., 2015*. Specifically, the region of contact between the two apposed objects was outlined

(on the spine head side) in each individual section using the arealist brush suitably set at 1 pixel size. In this way, a quasi two-dimensional sheet-like object representing the interfaced region was created along the z dimension. The total surface area was calculated by computing the smoothed upper bound surface, according to the formula

$$\text{Smoothed upper bound surface} = \sum_{k=0}^{n} \left( P_s\left(a\right) \times \tfrac{1}{2}T \right) + \left( P_s\left(b\right) \times \tfrac{1}{2}T \right) + \left[ A\left(a\right) - A\left(b\right) \right]$$

where n is the number of sections, a and b are the traced elements at the top and bottom of a section k of thickness T, Ps is the smoothed perimeter, and A is the area (*Cardona et al., 2012*). The area of the traced element in the section k=1 and in the section k=n were then subtracted from the smoothed upper bound surface value and the result was divided by 2 to get an approximate value of the apposed surface (AS). In oblique spines the ASI was not segmented because these spines were oriented obliquely or orthogonally to the cutting plane (~4.3% in each stack, across all groups; *Table 2*). The presence of the following structures was recorded for each spine: spine apparatus, spinula/s in the head or neck of the spine, mitochondria in the presynaptic element, tubules and cisterns of the SER, and components of the non-SER classified according to *Cooney et al., 2002*, including small uncoated vesicles, large coated or uncoated vesicles, and multivesicular bodies.

## Experimental design and statistical analysis

Statistical analysis was performed using LME models that include both random and fixed effects (*Laird and Ware, 1982*). The use of LME models was preferred over traditional ANOVA methods for several reasons. One reason is the ability to handle unbalanced data (e.g., differing numbers of dendrites sampled from each mouse, differing numbers of synapses measured from each dendrite). Another benefit of LME models, highly relevant for the current study, is the ability to measure covariates at different levels within the model; for example, ASI is measured at the level of individual synapses while spine density is measured at the level of dendrites.

The general matrix form of the LME is:

$$y = Zu + X\beta + \epsilon$$

where

$$u \sim N\left(0, \Sigma\right),$$

and

$$\epsilon \sim N\left(0, I\sigma^2\right).$$

In this model, y is the vector of response variables, u is a vector of random effects (independent and normally distributed with mean zero and covariance $\Sigma$), and β is the vector of fixed effects. Design matrices Z and X link the response variables to the random and fixed effects, and $\epsilon$ is the residual error, assumed to be independent and normally distributed with constant variance $\sigma^2$.

Two different model structures were used for the analyses. For synapse-level analysis (e.g., the response variable is the ASI of an individual synapse), we include condition and dendrite diameter as fixed effects and mouse and dendrite as random effects. For dendrite-level analyses (e.g., the response variable is the density of spines with a synapse on a dendrite), we include condition as a fixed effect and mouse as a random effect. In both cases, we test the hypothesis that there is no effect of condition by fitting a reduced model with the fixed effect of condition removed and performing an asymptotic likelihood ratio test to compare the full and reduced models. Parameter estimation of LME models was performed using numerical maximum likelihood estimations, implemented in R by the lmer() function of the lme4 package (*Bates et al., 2015*). If a significant effect of condition was identified, post hoc tests were performed using the glht() function of the multcomp package in R, with p-values adjusted for multiple comparisons using the single-step method (*Bretz et al., 2011*). The details of all LME models are included in *Tables 1, 3, and 4*.

## Additional information

### Funding

| Funder | Grant reference number | Author |
|---|---|---|
| NIH Office of the Director | DP 1OD579 (GT) | Giulio Tononi |
| National Institute of Mental Health | 1R01MH091326 (GT) | Giulio Tononi |
| National Institute of Mental Health | 1R01MH099231 (CC) | Giulio Tononi<br>Chiara Cirelli |
| National Institute of Mental Health | 1R01MH099231 (GT) | Giulio Tononi<br>Chiara Cirelli |
| National Institute of Neurological Disorders and Stroke | 1P01NS083514 (GT) | Giulio Tononi<br>Chiara Cirelli |
| National Institute of Neurological Disorders and Stroke | 1P01NS083514 (CC) | Giulio Tononi<br>Chiara Cirelli |
| U.S. Department of Defense | W911NF1910280 (CC) | Giulio Tononi<br>Chiara Cirelli |
| U.S. Department of Defense | W911NF1910280 (GT) | Giulio Tononi<br>Chiara Cirelli |

The funders had no role in study design, data collection and interpretation, or the decision to submit the work for publication.

### Author contributions

Sophia S Loschky, Giovanna Maria Spano, Andrea Schroeder, Kelsey Marie Nemec, Shannon Sandra Schiereck, Sebastian Weyn Banningh, Investigation; William Marshall, Formal analysis; Luisa de Vivo, Michele Bellesi, Resources; Giulio Tononi, Conceptualization, Writing - original draft, Writing - review and editing; Chiara Cirelli, Conceptualization, Supervision, Investigation, Writing - original draft, Writing - review and editing

### Author ORCIDs

Giovanna Maria Spano http://orcid.org/0000-0003-2626-754X
Luisa de Vivo http://orcid.org/0000-0002-5676-9279
Giulio Tononi http://orcid.org/0000-0002-3892-4087
Chiara Cirelli http://orcid.org/0000-0003-2563-677X

### Ethics

All animal procedures followed the National Institutes of Health Guide for the Care and Use of Laboratory Animals, and facilities were reviewed and approved by the IACUC of the University of Wisconsin-Madison and were inspected and accredited by AAALAC (animal protocol M005697).

### Decision letter and Author response

Decision letter https://doi.org/10.7554/eLife.84199.sa1
Author response https://doi.org/10.7554/eLife.84199.sa2

## Additional files

### Supplementary files
- MDAR checklist
- Source data 1. Dendrite data used for *Figures 3 and 4*.
- Source data 2. Synapse data used for *Figure 5*.

Data availability

All data generated in this study are included in the manuscript and supporting files (source data file 1 for figures 3 and 4; source data file 2 for figure 5).

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
