## [Editor Report]

This study provides compelling structural evidence on regulation of cerebellar synapses by sleep-wake states. The authors used serial block face scanning electron microscopy to obtain 3D reconstruction of more than 7,000 spines and their parallel fiber synapses in the mouse posterior vermis. The analysis shows that sleep increases the fraction of the 'naked' spines that don't carry a presynaptic partner at Purkinje cells and the authors propose that sleep promotes the pruning of branched synapses to single spines. This is an elegant and thorough study and the observations are important in light of the circuit-specific mechanisms by which sleep modulate synaptic structure and function.

---

## [Decision Letter]

**Decision letter after peer review:**

Thank you for submitting your article "Ultrastructural effects of sleep and wake on the parallel fiber synapses of the cerebellum" for consideration by *eLife*. Your article has been reviewed by 2 peer reviewers, and the evaluation has been overseen by a Reviewing Editor and Lu Chen as the Senior Editor. The reviewers have opted to remain anonymous.

Essential revisions:

1. The potential link between structural synaptic plasticity at the naked spines and its role in synaptic transmission at parallel fiber synapses is elusive. Functional data would suggest whether cerebellum has a different solution for sleep-induced synaptic depression. In the absence of data, the authors should at least discuss the existing literature on how sleep affects synaptic transmission at parallel fiber synapses.

2. Some of the animals were used in previous work. Do the results correlate in the same individual animal? Is it a correlation between sleep duration / quality and the number of naked synapses? For example, do mice that sleep 6 hours had less naked synapses then mice that sleep 8 hours? Does more NREM sleep time is correlated with less branched spines?

3. If behavioral and physiological results related to the amount and quality of sleep / wake prior to animal sampling are available, they should be explained and presented.

4. In this work, changes in the number of synapses, but not their size, were observed in the cerebellum. In contrast, changes in synapse size was found in the cortex and hippocampus of the same mice in previous publications of the authors. The circuit-specific effects of sleep should be discussed. Moreover, the role of Bergmann glial cells in sleep-induced synaptic plasticity should be discussed.

5. The abstract could be improved. It might be easier to follow if the authors compare the finding in the cerebellum to previous work in cortex and hippocampus.

6. A model diagram explaining the proposed changes in 'naked' spines during the sleep-wake cycle and the proposed functional consequences would be beneficial for the readers.

*Reviewer #1 (Recommendations for the authors):*

1. It is well established that sleep regulates synaptic plasticity. Specifically, this group already used scanning electron microscopy and described the effect of sleep and wake on the size and number of synapses in the cerebral cortex and hippocampus. In this study, these changes were not found in the cerebellum, however sleep and wake mildly affected the number of spine without synapses. Considering the previous findings and the relatively minor effect of sleep on synaptic plasticity found in this work, the novelty and significance of this work should be better explained. Is it the new brain region (cerebellum)?

2. Three groups of animals (wake, sleep and sleep deprived) and two time points are not sufficient to exclude clock effect and support sleep homeostatic process. Experiments under constant light and dark conditions, or in clock gene mutant mice, or sampling the mice in more than two time points (adding at least one more consecutive day or night group), would strengthen the conclusion.

3. In the introduction, it is recommended to introduce other potential functions for sleep (none synaptic), and the various evidences for global and circuit-specific sleep-dependent synaptic changes in humans, rodents and other animal models.

4. The potential link between cerebellar synaptic plasticity in the naked spines and a beneficial effect of sleep to motor learning is speculative. There is no functional data, only structural, and the effect on synaptic efficacy is unknown. Supportive behavioral and functional data could help to justify this hypothesis.

5. Some of the animals were used in previous work. Do the results correlate in the same individual animal? Can you find correlation between sleep duration and quality and the number of naked synapses? For example, do mice that sleep 6 hours had less naked synapses then mice that sleep 8 hours? Does more NREM sleep time is correlated with less branched spines?

6. Do behavioral and physiological data (EEG?) related to the amount and quality of sleep and wake prior to animal sampling is available? This should be explained and ideally presented.

6. In this work, changes in the number of synapses, but not their size, was observed in the cerebellum. In contrast, changes in synapse size was found in the cortex and hippocampus of the same mice (in previous publications of the same group). The circuit-specific effect of sleep should be discussed.

*Reviewer #2 (Recommendations for the authors):*

Include a model figure linking observations to the function of cerebellum.

Include data and/or a discussion of the role of Bergmann glial cells.

---

## [Author Response]

Essential revisions:1. The potential link between structural synaptic plasticity at the naked spines and its role in synaptic transmission at parallel fiber synapses is elusive. Functional data would suggest whether cerebellum has a different solution for sleep-induced synaptic depression. In the absence of data, the authors should at least discuss the existing literature on how sleep affects synaptic transmission at parallel fiber synapses.

To our knowledge, very little is known about the effects of sleep on synaptic transmission at the parallel fiber synapses. As pointed out in a recent cited review on the topic “The sleeping cerebellum” by Canto and colleagues (TINS 2017), “The Cerebellum Is Still an ‘Uncharted Land’ in Sleep Research”. We have, as much as possible, expanded the section in the conclusions that describes what is known about sleep/wake changes in spontaneous and evoked activity of Purkinje cells, as follows:

“Very little is known about the effects of sleep and wake on cerebellar physiology (Canto et al., 2017), and the available evidence is mostly confined to the Purkinje cells. Both their simple spike activity, which is driven by the excitatory inputs from parallel fiber synapses, and their complex spike activity, which reflects the excitatory input from a single climbing fiber, are lower during non-rapid eye movement (NREM) sleep than during wake or REM sleep (Mano, 1970; Marchesi and Strata, 1971; Hobson and McCarley, 1972; Harlay et al., 1974; Canto et al., 2017). The response of Purkinje cells to microiontophoretically applied glutamate is also smaller in NREM sleep than in wake (Andre and Arrighi, 2001). This effect is not accounted for by changes in spontaneous firing rate but may be associated with the decreased level of acetylcholine and noradrenaline in NREM sleep, which is observed across the brain, including in the cerebellum (Andre and Arrighi, 2001). However, whether the sleep-wake cycle is also accompanied by changes in the number or strength of synaptic connections was completely unknown.”

2. Some of the animals were used in previous work. Do the results correlate in the same individual animal? Is it a correlation between sleep duration / quality and the number of naked synapses? For example, do mice that sleep 6 hours had less naked synapses then mice that sleep 8 hours? Does more NREM sleep time is correlated with less branched spines?

We have performed two analyses. First, we have compared the results in the same individual animal in CA1 (mean ASI) and cerebellum (branched synapses and proportion of spines without a synapse), because these 2 studies shared the same 14 mice (only 8 mice are shared between cerebellum and cortex). While correlations are generally in the "right direction" (e.g. mice with larger ASI tend to have higher density of branched synapses), there are so few data points that nothing is significant. Because we can't draw any strong conclusions, we prefer not to include these results in the main text. Examples of the “best” correlations are included in Author response image 1.

**Author response image 1. sa2fig1:** 

In the second analysis (Cb only) we tested whether the proportion of spines lacking a synapse correlates with the % of sleep in the last 6-8hrs, and found no correlation (r = -0.13, p = 0.809). This is not surprising, since we specifically used strict sleep criteria to select the mice (sacrifice occurred after >45 min of sleep, interrupted by periods of wake lasting less than 4 min, and after spending at least 75% of the previous 6–8 hours asleep). As stated, to avoid possible tissue damage due to the EEG electrodes, sleep/wake were measured based on rest/activity and video recordings; we do not have a reliable way to measure NREM and REM sleep separately.

3. If behavioral and physiological results related to the amount and quality of sleep / wake prior to animal sampling are available, they should be explained and presented.

We don’t have additional behavioral data (in addition to the % of sleep and wake) in the last 6-8 hours, nor other physiological data. As stated, to avoid possible tissue damage due to the EEG electrodes, sleep/wake were measured based on rest/activity and video recordings; we do not have a reliable way to measure NREM and REM sleep separately.

4. In this work, changes in the number of synapses, but not their size, were observed in the cerebellum. In contrast, changes in synapse size was found in the cortex and hippocampus of the same mice in previous publications of the authors. The circuit-specific effects of sleep should be discussed. Moreover, the role of Bergmann glial cells in sleep-induced synaptic plasticity should be discussed.

1. We have added the following section to the discussion:

“We also found that sleep and wake do not affect the size of the ASI, a structural measure of synaptic strength (Desmond and Levy, 1988; Buchs and Muller, 1996; Cheetham et al., 2014). This is in contrast with our findings in cortex and hippocampus, where sleep and wake lead to changes in the average strength of the synapses rather than in their number (Cirelli and Tononi, 2021). Thus, the effects of sleep and wake point in the same direction in all three brain regions that we have examined so far, with more and/or stronger synapses after wake than after sleep. However, the underlying mechanisms are quite distinct. In the cerebellum sleep and wake affect the number of branched synapses and the relative number of spines that carry a synapse but not the average synaptic strength. In cerebral cortex and hippocampus instead, sleep and wake mainly affect the size of the ASI but not the number of synapses. These regional differences are notable, since the current study utilized the same 14 mice previously used for the analysis of CA1 (Spano et al., 2019) and 8 of them were also used for the analysis of the primary motor and sensory cortex (de Vivo et al., 2017), but they are not unexpected. Specifically, the lack of changes in ASI size is less surprising if one considers the striking difference in the distribution of synaptic sizes between cortex and hippocampus on one hand, and the cerebellum on the other hand. In the first two regions the distributions are log-normal and bimodal, respectively, meaning that synaptic sizes (and strengths) cover a wide range and include large synapses. By contrast, the sizes of the parallel fiber synapses follow a unimodal distribution, meaning that these synapses tend to be all of the same small-medium size. This narrow range of sizes strongly suggests that, in general, plastic changes in these synapses are implemented more via all or none changes (adding or removing a synapse) than through graded changes in size.”

2. We have added the following section in the conclusions about Bergmann glia:

“In the cerebral cortex, we previously reported that the sleep/wake changes in ASI size are accompanied by several changes in the peripheral astrocytic processes surrounding many cortical synapses. Specifically, these processes contain more numerous but smaller glycogen granules after spontaneous and forced wake compared to sleep, suggesting increased glycogen turnover to meet the high energy demand of wake (Bellesi et al., 2018). Moreover, after a few hours of sleep deprivation astrocytic processes get closer to the synaptic cleft, likely reflecting an increased need for glutamate clearance (Bellesi et al., 2015), and astrocytic phagocytosis of synaptic elements is increased (Bellesi et al., 2017). Bergmann glia, the astrocytic cells of the cerebellum, establish extensive contacts with parallel fiber synapses and their distal processes respond to the electrical stimulation of the parallel fibers with localized increases in calcium levels (Grosche et al., 1999). Calcium signaling in Bergmann glia can, in turn, modulate the activity of neighboring Purkinje cells (Wang et al., 2012), in line with strong neuron-glia crosstalk. Moreover, contrary to cortical astrocytes, Bergmann glia express AMPA receptors, which are required for the normal development of the parallel fiber synapses (Saab et al., 2012). Bergmann glia is also routinely involved in the phagocytosis of synaptic and extrasynaptic elements, a process that increases after motor learning (Morizawa et al., 2022). Thus, given the multiple ways Bergmann glia can modulate cerebellar activity (De Zeeuw and Hoogland, 2015), and specifically the parallel fiber synapses that are the focus of the current study, future experiments should test whether sleep and wake affect the structure and function of these cells.”

5. The abstract could be improved. It might be easier to follow if the authors compare the finding in the cerebellum to previous work in cortex and hippocampus.

We have revised the abstract according to the suggestion.

“Multiple evidence in rodents shows that the strength of excitatory synapses in the cerebral cortex and hippocampus is greater after wake than after sleep. The widespread synaptic weakening afforded by sleep is believed to keep the cost of synaptic activity under control, promote memory consolidation, and prevent synaptic saturation, thus preserving the brain’s ability to learn day after day. The cerebellum is highly plastic and the Purkinje cells, *the sole output neurons of the cerebellar cortex, are endowed with* a staggering number of excitatory parallel fiber synapses. However, whether these synapses are affected by sleep and wake is unknown. Here we used serial block face scanning electron microscopy to obtain the full 3D reconstruction of more than 7,000 spines and their parallel fiber synapses in the mouse posterior vermis. This analysis was done in mice whose cortical and hippocampal synapses were previously measured, revealing that average synaptic size was lower after sleep compared to wake with no major changes in synapse number. Here, instead, we find that while the average size of parallel fiber synapses does not change, the number of branched synapses is reduced in half after sleep compared to wake, corresponding to ~16% of all spines after wake and ~8% after sleep. Branched synapses are harbored by two or more spines sharing the same neck and, as also shown here, are almost always contacted by different parallel fibers. These findings suggest that during wake, coincidences of firing over parallel fibers may translate into the formation of synapses converging on the same branched spine, which may be especially effective in driving Purkinje cells to fire. By contrast, sleep may promote the off-line pruning of branched synapses that were formed due to spurious coincidences.”

6. A model diagram explaining the proposed changes in 'naked' spines during the sleep-wake cycle and the proposed functional consequences would be beneficial for the readers.

We have added a model diagram (Figure 6) summarizing the main findings and their putative functional consequences.

Reviewer #1 (Recommendations for the authors):1. It is well established that sleep regulates synaptic plasticity. Specifically, this group already used scanning electron microscopy and described the effect of sleep and wake on the size and number of synapses in the cerebral cortex and hippocampus. In this study, these changes were not found in the cerebellum, however sleep and wake mildly affected the number of spine without synapses. Considering the previous findings and the relatively minor effect of sleep on synaptic plasticity found in this work, the novelty and significance of this work should be better explained. Is it the new brain region (cerebellum)?

The novelty and significance come from the focus on a type of synapse, the parallel fiber synapse of the cerebellum, which is the most abundant type of excitatory synapse in the mammalian brain but had never been studied across the sleep/wake cycle before. We have tried to further clarify this point in the introduction. Given the exorbitant number of parallel fiber synapses, an increase in the proportion of spines without such synapses from 5 to 10% after a few hours of sleep should perhaps not be considered that “mild.”

“However, very little if anything is known about how sleep and wake modulate cerebellar plasticity in general (Canto et al., 2017) and, more specifically, whether they affect the number and strength of the parallel fiber synapses.”

2. Three groups of animals (wake, sleep and sleep deprived) and two time points are not sufficient to exclude clock effect and support sleep homeostatic process. Experiments under constant light and dark conditions, or in clock gene mutant mice, or sampling the mice in more than two time points (adding at least one more consecutive day or night group), would strengthen the conclusion.

This is a good point but we actually never claimed that using 3 groups is sufficient to rule out clock effects. As stated, the use of 3 groups allows us to compare the sleep group with two wake groups, extended wake during the day and spontaneous wake at night, thus controlling for time-of-day effects (day vs. night) and the possible “stress” effects of sleep deprivation (spontaneous vs. forced wake). We have added a sentence in the experimental design section stating that while the differences between S and EW mice cannot be accounted for by circadian factors, additional experiments under constant light and dark conditions, or in clock gene mutant mice, would be necessary to rule out clock effects when comparing S and W mice.

3. In the introduction, it is recommended to introduce other potential functions for sleep (none synaptic), and the various evidences for global and circuit-specific sleep-dependent synaptic changes in humans, rodents and other animal models.

We agree that there are several other potential functions of sleep, and we have covered this topic in recent reviews (e.g. sleep to replenish and detoxify, in Cirelli and Tononi, Semin Cell Dev Biol 2022). However, the introduction reflects the scope of the current study, which was designed to address a very specific question related to changes in synaptic strength in the cerebellum.

4. The potential link between cerebellar synaptic plasticity in the naked spines and a beneficial effect of sleep to motor learning is speculative. There is no functional data, only structural, and the effect on synaptic efficacy is unknown. Supportive behavioral and functional data could help to justify this hypothesis.

We agree that the suggested functional consequences of the ultrastructural changes are speculative (although structure and function are correlated), and we have been careful in presenting them as “hypothesis”. We now have added a final paragraph to the conclusions to further stress this point. We note, however, that we discuss at length the functional data that are at least consistent with our interpretation, including the electrophysiological evidence for many silent parallel fiber synapses and the modeling studies showing that, given the huge number of these synapses, the most effective way to store information may be to make only some of them effective (eloquent).

“While speculative, this interpretation is consistent with electrophysiological, anatomical, and modeling studies showing that (1) many parallel fiber synapses are silent and (2) given the exorbitant number of these synapses, information storage is promoted when only some of them are effective (eloquent).”

5. Some of the animals were used in previous work. Do the results correlate in the same individual animal? Can you find correlation between sleep duration and quality and the number of naked synapses? For example, do mice that sleep 6 hours had less naked synapses then mice that sleep 8 hours? Does more NREM sleep time is correlated with less branched spines?

Please see above, Essential revisions, Point 2.

6. Do behavioral and physiological data (EEG?) related to the amount and quality of sleep and wake prior to animal sampling is available? This should be explained and ideally presented.

Please see above, Essential revisions, Point 3.

6. In this work, changes in the number of synapses, but not their size, was observed in the cerebellum. In contrast, changes in synapse size was found in the cortex and hippocampus of the same mice (in previous publications of the same group). The circuit-specific effect of sleep should be discussed.

Thank you for this suggestion. Please see above, Essential revisions, Point 4 (1).

Reviewer #2 (Recommendations for the authors):Include a model figure linking observations to the function of cerebellum.

Thank you for this suggestion. Please see above, Essential revisions, Point 6.

Include data and/or a discussion of the role of Bergmann glial cells.

Thank you for this suggestion. Please see above, Essential revisions, Point 4 (2).